# BMP-9 Improves the Osteogenic Differentiation Ability over BMP-2 through p53 Signaling In Vitro in Human Periosteum-Derived Cells

**DOI:** 10.3390/ijms242015252

**Published:** 2023-10-17

**Authors:** Jin-Ho Park, Eun-Byeol Koh, Young-Jin Seo, Hye-Seong Oh, June-Ho Byun

**Affiliations:** 1Department of Nutritional Science, University of Michigan School of Public Health, Ann Arbor, MI 48109, USA; 2Department of Oral and Maxillofacial Surgery, Gyeongsang National University School of Medicine, Gyeongsang National University Hospital, Institute of Medical Sciences, Gyeongsang National University, Jinju 52727, Republic of Korea; 3Department of Convergence Medical Science, Gyeongsang National University, Jinju 52828, Republic of Korea

**Keywords:** BMP-9, p53, PI3K/AKT, periosteum-derived cells, osteogenic differentiation

## Abstract

Bone morphogenetic proteins (BMPs) have tremendous therapeutic potential regarding the treatment of bone and musculoskeletal disorders due to their osteo-inductive ability. More than twenty BMPs have been identified in the human body with various functions, such as embryonic development, skeleton genesis, hematopoiesis, and neurogenesis. BMPs can induce the differentiation of MSCs into the osteoblast lineage and promote the proliferation of osteoblasts and chondrocytes. BMP signaling is also involved in tissue remodeling and regeneration processes to maintain homeostasis in adults. In particular, growth factors, such as BMP-2 and BMP-7, have already been approved and are being used as treatments, but it is unclear as to whether they are the most potent BMPs that induce bone formation. According to recent studies, BMP-9 is known to be the most potent inducer of the osteogenic differentiation of mesenchymal stem cells, both in vitro and in vivo. However, its exact role in the skeletal system is still unclear. In addition, research results suggest that the molecular mechanism of BMP-9-mediated bone formation is also different from the previously known BMP family, suggesting that research on signaling pathways related to BMP-9-mediated bone formation is actively being conducted. In this study, we performed a phosphorylation array to investigate the signaling mechanism of BMP-9 compared with BMP-2, another influential bone-forming growth factor, and we compared the downstream signaling system. We present a mechanism for the signal transduction of BMP-9, focusing on the previously known pathway and the p53 factor, which is relatively upregulated compared with BMP-2.

## 1. Introduction

The bone provides the body with structural integrity, it promotes movement, protects vital organs, and is the sheath for bone marrow, which promotes blood cell synthesis and the metabolism of essential minerals, such as phosphorus and calcium [1]. Two key steps in bone regeneration are osteo-induction (in which mesenchymal stem cells become bone-forming osteoblasts) and differentiating between these osteoblasts to produce the osteogenic matrix. BMPs play an important role in regulating these two steps of bone regeneration [2]. Due to their osteo-inductive abilities, BMPs have tremendous therapeutic potential for treating bone and musculoskeletal disorders. BMPs are growth factors with multiple functions, and they belong to the TGF-β family. More than 20 BMPs have been identified in the human body, and they carry out functions such as embryogenesis, skeletal formation, hematopoiesis, and neurogenesis [3]. BMPs can induce the differentiation of MSCs into the osteoblast lineage, and they promote the proliferation of osteoblasts and chondrocytes. In human bone regeneration, cell division and differentiation are regulated by several growth factors. BMP signaling is also involved in tissue remodeling and the regeneration processes which maintain adult homeostasis [3,4].

BMPs are used in orthopedic applications, such as spinal fusions, non-unions, and oral surgery. Most BMPs have been shown to induce osteogenesis in vivo [5,6], and both BMP-2 and BMP-7 have been tested in clinical trials [7,8,9]. BMP-2 and BMP-7 are approved by the FDA [10]; BMP-2 causes more overgrown bone than any other BMP, so it is used more widely in clinical settings [11]. However, it is unclear as to whether BMP-2 and -7 are the most potent BMPs to induce osteogenesis. Recent studies have shown that BMP-9 is the most potent inducer of the osteogenic differentiation of mesenchymal stem cells, both in vitro and in vivo, although its precise role in the skeletal system is unclear [12,13]. In comparison with other BMP family members, BMP-9 has a slightly different form of ligand–receptor interaction. BMP-9 binds to ALK1 and ALK2 type I receptors with a high degree of affinity, whereas the signal transduction of BMP-2/4/6/7 is mainly mediated through ALK3 and ALK6. In addition, the molecular mechanisms of BMP-9-mediated bone formation are also different from other BMP families. Increasing evidence suggests that BMP-9-mediated osteogenesis occurs through distinct signaling pathways that differ from other BMPs. Another notable feature of BMP-9 is that it binds with a high degree of affinity to the BMP co-receptor, endoglin [12,14]. Endoglin has been shown to activate Akt in endothelial cells by interacting with PI3K p110-a and p85 subunits via GIPC1 [15]. A recent study revealed that the receptor for BMP-9 is a complex composed of ALK1 and endoglin rather than a type II receptor, suggesting a new and unique aspect of the BMP-9 signaling mechanism [16]. For example, noggin is an antagonist of BMPs, which usually inhibits the BMP-2/4/7 signaling pathways, but it does not inhibit the BMP-9 signaling pathway. Moreover, BMP-3, a generally recognized inhibitor of BMPs, can inhibit BMP-2/4/6/7-mediated osteogenesis, but its inhibitory effect on BMP-9-mediated osteogenesis is very weak [17,18]. This indicates that BMP-9 can induce osteogenesis by following a pathway that is different from the traditional BMP signaling pathway in order to fulfill its role, or to bypass pathways to regulate the expression of downstream genes. Nevertheless, details of intracellular signaling, transduced with BMP-9, remain obscure [19].

Protein phosphorylation is an important signaling mechanism in cells and tissues, and it affects the cell signaling system through various pathways. Here, we performed a phospho–kinase array to comparatively investigate the signaling mechanism of BMP-9 compared with BMP-2 (another potent bone-forming cytokine) in osteoblasts. Through the phospho–kinase array, p53 was identified as the most significantly upregulated protein in the BMP-9-treated group. p53 is frequently cited as a negative regulator in bone differentiation [20], but it also acts as a transcriptional activator for many bone-specific genes. Currently, there is controversy surrounding the osteoblast differentiation of p53 [21,22]. As such, although p53 inhibits osteoblast proliferation and promotes osteoblast apoptosis to reduce bone formation, it also positively affects osteoblast differentiation through the Akt-FoxOs pathway [23].

Therefore, we investigated the roles and pathways of p53, following BMP-9 treatment, during osteoblast differentiation. Although BMP-9 has been studied extensively in relation to osteogenesis, there is much controversy about its mechanism of action. This study aimed to evaluate the effect of BMP-9 on osteogenesis in human periosteum-derived cells (hPDCs) compared with the BMP-2 factor, and to investigate the most likely relevant cell signaling mechanisms.

## 2. Results

### 2.1. Osteogenic Potential of hPDCs Induced by BMP-9 In Vitro

hPDCs were cultured by adding BMP-2 and BMP-9 to osteogenic induction media (OM) at doses of 1, 10, and 100 ng/mL, respectively. After 1 week of culturing, ALP was visualized via staining with NBT/BCIP (Figure 1A), and ALP was quantified by measuring the absorbance after treatment with pNPP (Figure 1B). Compared with the group treated with only OM, the phenotype and activity of ALP increased as the concentration of BMP-2 increased in the group treated with BMP-2, and the phenotype and activity of ALP were the highest in the group treated with 100 ng/mL BMP-9. After 3 weeks of culturing, cells were stained with Alizarin red S to visualize the phenotype for calcium, a late marker of bone differentiation. To quantify the degree of staining, the cells were dissolved in cetylpyridinium chloride and quantified (Figure 1C). Similar to the ALP results, calcium deposition was the highest in the group treated with 100 ng/mL of BMP-9 (Figure 1D). BMP-2 and BMP-9 induced bone differentiation most significantly at 100 ng/mL, therefore, the treatment concentration of BMP was fixed at 100 ng/mL in subsequent experiments. These results indicate that BMP-9, along with BMP-2 (a representative factor among BMP family members), also upregulates the osteogenic differentiation of hPDCs in a dose-dependent manner. Nevertheless, the induction effect of BMP-9 is clearly better than that of BMP-2.

### 2.2. Comparison of Intracellular Signaling Pathway Differences in BMP-Induced hPDCs

To compare intracellular bone differentiation signaling pathways caused by BMP, the group differentiated with OM was used as a control group, and BMP-2 and BMP-9 were added during osteogenic differentiation, respectively. Every group was cultured for 1 week to perform a phosphorylation array. In the BMP-9 group, six proteins with higher phosphorylation levels than in the BMP-2 group were selected. p53 (S392, S15, S46), β-catenin, PRAS40 (T246), and PLC-γ1 (Y783) were higher in the BMP-9 group than in the BMP-2 group, and in particular, p53 (S392) showed the clearest difference (Figure 2A,B). On the other hand, it was confirmed that phosphorylation forms of JNK and p38, which are MAPKs, were lower in the BMP-9 group than in the BMP-2 group in the phosphorylation array. It is very significant that MAPK, which is known to be mainly involved in the osteogenic differentiation process, was reduced, and this is considered a very impressive result that can support the hypothesis that BMP-9 has a signaling system other than general differentiation signaling.

Based on the phosphorylation array results, additional Western blotting was performed to confirm that the p53 group collectively increased due to Wnt signaling and BMP-9, which are generally well known to have a relationship with BMP. As a result, in the BMP-9-treated osteoblast differentiation group, most of the downstream factors of the Wnt signal were downregulated or they exhibited no change (Figure 3A,B). On the other hand, in the p53 group, it was confirmed that the factors increased significantly once again, similar to the array results (Figure 3C,D). This suggests that BMP-9, which, although it plays a positive role in osteoblast differentiation, does not participate in the Wnt signaling pathway; rather, downregulates it and that BMP-9 signaling is directly or indirectly closely related to p53.

### 2.3. Reduction in Osteogenic Differentiation by Pifithrin-α

As a result of ALP staining, after 1 week of culturing (Figure 4A,B), the level of expression was reduced in a dose-dependent manner, and it was confirmed that ALP staining faded significantly, starting with the Pifithrin-α (Pfα) 0.1 uM group. Moreover, ALP activity also showed the same pattern. After 1 week of culturing, as a result of ALP staining (Figure 4A), the level of expression was reduced in a dose-dependent manner, and it was confirmed that ALP staining faded significantly, starting with the Pfα 0.1 uM group.; ALP activity also showed the same pattern (Figure 4B). qRT-PCR was performed to directly confirm the expression level of factors involved in the early stage of differentiation and the overall osteogenic differentiation process at the mRNA level (Figure 4C). As a result of analyzing *Runx2* (an osteogenic differentiation transcription factor) in the first week of differentiation, it was confirmed that the expression level of *Runx2* was higher in the BMP-9 group than in the BMP-2 group, and it was downregulated due to treatment with Pfα. The relative expression levels of *Runx2*, *ALP*, and *BGLAP (Osteocalcin)* are indicated. The results of analyzing ALP (an early osteogenic differentiation transcription factor) in the second week of differentiation, and BGLAP (a late osteogenic differentiation transcription factor) in the fourth week of differentiation, showed the same pattern in terms of expression levels. This was finally re-verified through an experiment to confirm the degree of calcium deposition. BMP-9 clearly upregulated Alizarin red staining and the calcium content compared with the control group, but as p53 was inhibited, the degree of differentiation also consistently decreased.

### 2.4. Mechanism of Action of BMP-9 and p53

To confirm the action pathways of p53 and BMP-9, we checked the expression of the PI3K/Akt pathway, which is known to play a significant role, although there is still much controversy (Figure 5A,B). As a result, the expression of p-PI3K and its downstream signal, p-AKT, in the BMP-9 group significantly increased compared with the control and BMP-2 group. On the other hand, we confirmed that when p53 was inhibited using Pfα, it decreased again. In addition, an interesting result was observed in that MDM2, which has an antagonistic effect downstream of the PI3K/AKT pathway, showed the opposite effect. These results suggest that the activation of p53 with BMP-9 leads to the activation of its downstream factor, AKT, via the PI3K pathway. On the contrary, the suppression of p53 resulted in an elevation of MDM2, which exhibited an antagonistic influence. This observation suggests that the PI3K/AKT signaling pathway plays a significant role in the upregulation of osteogenic differentiation in hPDCs induced with BMP-9.

In addition, to confirm that the BMP function, *Noggin*, a factor expressed to regulate BMPs when they increase, was verified at the mRNA level (Figure 5C). It was confirmed that the expression level of Noggin was maintained at a high level due to treatment with BMP-9 from the first week of osteogenic differentiation to the fourth week. The micro-RNA groups (*miR-34a/b/c* and *miR-145*) that regulate p53 were also established through qPCR, and the results showed that *miR-34a/b* was significantly higher in the BMP-9-induced group than in the control OM group (Figure 5D). An apparent increase was observed, and this was likely to be primarily regulated with *miR-34a/b* using the p53 signal, which is involved in the role of BMP-9.

## 3. Discussion

Compared with other BMPs in the BMP family, the strong osteogenesis-inducing ability of BMP-9 has been demonstrated in several studies [18,24,25]. In several other studies, BMP-3, a negative osteogenesis regulator, effectively inhibited osteogenic differentiation induction with other BMPs, including BMP-2, 6, and 7, but it did not inhibit osteogenesis with BMP-9 [18]. In addition, Noggin, which is well known as a BMP antagonist, is known to not inhibit BMP-9 intracellular signal transduction, unlike the other BMPs [17]. Therefore, BMP-9 has a unique bone formation signal compared with other BMPs during the osteoblast differentiation of stem cells. However, the exact mechanism of this non-canonical pathway has not been fully elucidated [19].

We first treated hPDCs with BMP-2 and BMP-9 to investigate this mechanism to induce osteoblast differentiation. As described above, ALP staining, an activity assay (Alizarin red S staining), and a calcium content assay were used to evaluate the in vitro osteogenic phenotype of the hPDCs. It was verified that the degree of osteoblast differentiation was significantly increased in the BMP-9-induced group compared with the BMP-2-induced group, like other MSCs. In addition, after the differentiation of hPDCs into BMP-2 and BMP-9 groups for one week, intracellular signaling pathways during osteoblast differentiation were explored through phospho-kinase array experiments. It is well known that BMP can activate Runx2 through the MAPK signaling pathway [26,27]. However, the low phosphorylation of JNK and p38, which are MAPKs, was observed in the BMP-9 group, indicating that BMP-9 differentiates osteoblasts via pathways other than the MAPK pathway. At the same time, it was found that the phosphorylation of p53, a representative tumor suppressor through the induction of cell cycle arrest and apoptosis, was significantly increased in the BMP-9 treatment group. The Western blot assay confirmed that p53 increased in the BMP-9 group. Additionally, it was found that the p53 inhibitor reduced the degree of osteoblast differentiation in the BMP-9 group. This result suggests that p53 signaling plays an important role in the osteoblast differentiation of hPDCs. The representative tumor suppressor, p53, is a sequence-specific transcription factor that regulates the expression of various direct target genes involved in cell cycle arrest and in the induction of apoptosis [28,29]. p53 signaling is complex and regulated by a variety of factors. Among them, studies have recently shown that the PI3K/AKT/MDM2 axis regulates bone homeostasis by selectively activating many signal molecules through the PI3K/AKT pathway in bone, especially osteoblasts [30]. The present study revealed a correlation between the BMP-9 inhibition of the PI3K/Akt/MDM2 pathway, which is the pathway of p53 bone homeostasis, and an increase in p53 activity, which promotes the differentiation of osteoblasts [31,32,33,34,35,36]. In addition, miRNAs that regulate p53 transcription were identified. miRNA is a single-stranded non-coding RNA consisting of 22 nucleotides or fewer. It plays a role in regulating mRNA by degrading its target or inhibiting translation by annealing to the 3′ UTR of mRNA [37]. The post-transcriptional mechanisms mediated by numerous miRNAs are being actively studied in terms of their relation to osteoblast differentiation [38]. miRNAs are expected to bind to a large number of target mRNAs, and p53 is known to participate in various functions in normal cell physiology as they are regulated at the mRNA level with miRNAs. Therefore, we specifically confirmed the miRNA changes during the osteoblast differentiation process that were induced using BMP-9. miRNA-34a/34b/34c are targets of p53, they participate in cell proliferation and cell growth control, they increase in number during the osteoblast differentiation of rat cranial cells [39,40], and miRNA-145 is regulated by mutations in the p53 gene. It is known to be regulated [41]. The findings of this study suggest that the activation of p53, induced using BMP-9, is related to osteoblast differentiation.

This study confirmed that PI3K/Akt/MDM2, a signaling pathway of p53, was activated in hPDC, and induced using BMP-9. It was confirmed that PI3K/Akt/MDM2 axis activation promotes osteoblast differentiation through feedback action with p53. Using this mechanism, we suggest that p53 enhances osteogenic differentiation in hPDCs through the PI3K/Akt/MDM2 axis. However, since p53 plays various roles in cells and organisms, the findings of this study may be related to apoptosis and cell cycle arrest, and further research is needed. In conclusion, it is suggested that regulating specific signal transduction with BMP-9 can induce osteoblast differentiation via a new approach that is different from the existing mechanism. These findings imply that BMP-9 should be investigated for clinical applications such as extensive segmental bony defects, non-union fractures, and/or spinal fusions as an efficient bone regeneration agent, particularly in combination with adjuvant therapy.

## 4. Materials and Methods

### 4.1. Materials

The Human Phospho-Kinase Array Kit and recombinant human BMP-2/ BMP-9 were purchased from R&D Systems (Minneapolis, MN, USA). The primary antibodies against anti-human Phospho-p53 (Ser6, 9, 15, 46, 392, and Thr81), Wnt5a/b, LRP6, Phospho-LRP6, Dvl2, Dvl3, Axin1, GSK3β, Phospho-GSK3β, PI3K, Phospho-PI3K, AKT, Phospho-AKT, MDM2, and β-actin were purchased from Cell Signaling Technology (Danvers, MA, USA). The iScript cDNA Synthesis Kit was purchased from Bio-Rad Laboratories, Inc. (Hercules, CA, USA). All primers and probes (RUNX2 #Hs00231692_m1; ALPL #Hs010291444_m1; BGLAP #Hs01587814_g1; NOG #Hs00271352_s1; GAPDH #Hs99999905-m1) were purchased from Applied Biosystems, Inc. (Bedford, MA, USA). Moreover, the TaqMan MicroRNA Reverse Transcription Kit and TaqMan Universal Master mix were from Applied Biosystems, Inc.

### 4.2. Culture and Osteogenic Differentiation of the hPDCs

The hPDCs were harvested from mandibles during the surgical extraction of impacted lower third molars [42,43]. Periosteal tissues were washed and cultured at 37 °C, in air at 95% humidity, and in 5% CO_2_ in 100 mm culturing dishes containing DMEM supplemented with 10% heat-inactivated fetal bovine serum (FBS), 100 IU/mL penicillin, and 100 μg/mL streptomycin. Osteoblast differentiation was induced by culturing periosteal cells (seeded at a density of 5 × 10^4^ cells/well on 24-well plates) at passages 3 to 5 in an osteogenic induction medium composed of DMEM supplemented with 10% FBS, 50 µg/mL L-ascorbic acid 2-phosphate, 10 nM dexamethasone, and 10 mM β-glycerophosphate. Upon reaching 90% confluency, adherent cells were passaged via gentle trypsinization and reseeding in a fresh medium. BMP-2 or BMP-9 was applied at each concentration (1, 10, or 100 ng/mL) at the same time as the osteogenic differentiation period, and the cell culture medium was changed every 3 days.

### 4.3. Evaluation of Osteogenic Phenotype

The osteogenic phenotypes of hPDCs were assessed, as previously described [43,44]. ALP staining and activity assays were performed on day 10 of culturing, whereas alizarin red S staining and calcium contents were determined on day 20 of culturing. Media were changed every 3 days, and every factor was added at each medium change.

### 4.4. Phosphorylation Array

A Proteome Profiler membrane-based immunoassay was performed using the Human Phospho-Kinase Array Kit, in accordance with the manufacturer’s instructions, to measure the relative level of signal transduction of protein phosphorylation that changes in the osteogenic cells induced with BMP-2 or BMP-9.

### 4.5. Western Blot Analysis

The cells were placed in a RIPA buffer (Cell Signaling Technology, Danvers, MA, USA) containing a protease and phosphatase inhibitor cocktail to extract all proteins. After 20 min, the cell pellets were sonicated and centrifuged, and the supernatant was separated using sodium dodecyl sulfate-polyacrylamide gel electrophoresis (SDS-PAGE), which was then transferred onto a polyvinylidene difluoride membrane (Millipore, Burlington, MA, USA). The membranes were then probed with primary antibodies against anti-human Phospho-p53 (Ser6, 9, 15, 46, 392, and Thr81), Wnt5a/b, LRP6, Phospho-LRP6, Dvl2, Dvl3, Axin1, GSK3β, Phospho-GSK3β, PI3K, Phospho-PI3K, AKT, Phospho-AKT, MDM2, and β-actin, respectively. Specific antibody binding was detected using horseradish peroxidase-conjugated secondary antibodies, then, this was visualized using an enhanced chemiluminescence detection reagent (Invitrogen, Carlsbad, CA, USA).

### 4.6. Reverse Transcription and Quantitative RT-PCR (qRT-PCR) Using TaqMan Assays

The expression levels of osteoblast-related genes (Runx2, ALP, and BGLAP) and Noggin were analyzed in factors (BMP-2, BMP-9, and Pfα) treated with hPDCs at the indicated times. First-strand cDNA was generated using the random hexamer primers provided in the iScript cDNA Synthesis Kit. All primers and probes were amplified using a kit in accordance with the manufacturer’s instructions.

To measure the levels of *miR-34a*, *b*, *c*, and *miR-145*, reverse transcription was performed using the TaqMan MicroRNA Reverse Transcription Kit with specific stem-loop primers for each miRNA. The quantitative polymerase chain reaction was then performed using the TaqMan Universal Master mix and the following TaqMan micro RNA assays (Applied Biosystems): *hsa-miR-34a* (#000426), *hsa-miR-34b* (#000427), *hsa-miR-34c* (#000428), and *hsa-miR-145* (#002278). As a control small RNA, U6 small nucleolar RNA (#001973) was used. The amplification conditions were as follows: 50 °C for 2 min, 95 °C for 10 min, then 40 cycles of 94 °C for 15 s, and 60 °C for 1 min in 96-well plates using the ViiA™ 7 RealTime PCR System (Applied Biosystems). As an internal control, glyceraldehyde 3phosphate dehydrogenase (GAPDH) was used. Every experiment was carried out in triplicate.

### 4.7. Statistical Analysis

Each experiment was repeated at least three times in a row. As representative data, the results of one of three independent experiments are shown. The mean and standard deviation are used to express data. GraphPad Prism software (v.7.0, GraphPad Software Inc., San Diego, CA, USA) was used for statistical analyses. For all statistical analyses, we used one-way ANOVA with Tukey’s multiple comparisons (v.22.0, SPSS Statistics 22 software, IBM, Armonk, NY, USA). Comparisons with *p* < 0.05 were considered statistically significant.

## Figures and Tables

**Figure 1 ijms-24-15252-f001:**
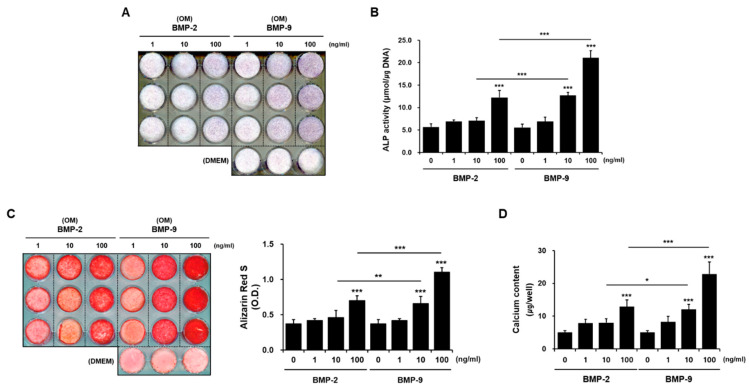
Comparison of osteogenesis induction using doses of BMP-2 and BMP-9. BMP factors play a beneficial role in the osteoblast differentiation of hPDC by improving ALP activity (**A**,**B**), alizarin red positive mineralization (**C**), and the amount of calcium deposited (**D**) in a dose-dependent manner. However, as the dose of BMP-9 increases, the degree of osteogenic differentiation is further improved compared with BMP-2. OM; osteogenic induction media (*n* = 3; * *p* < 0.05, ** *p* < 0.01, *** *p* < 0.001).

**Figure 2 ijms-24-15252-f002:**
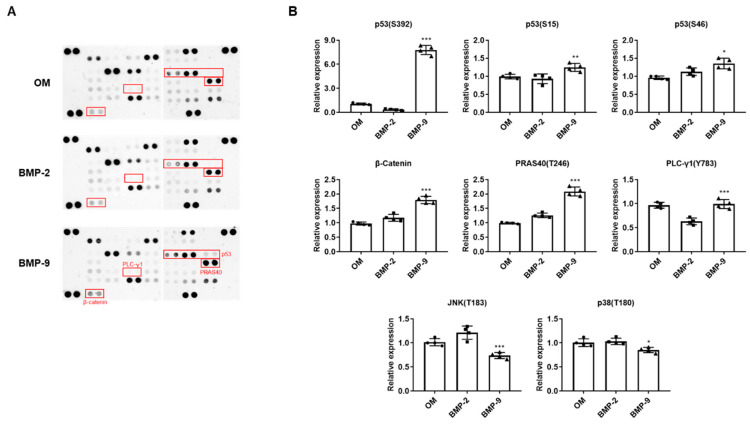
Comparison of phosphorylation arrays according to osteogenic differentiation using BMP-2 and BMP-9. The membrane array results compare changes in protein phosphorylation due to BMPs (**A**), and through quantification, factors that were significantly upregulated or downregulated due to BMP-9 were selected (**B**). OM; osteogenic induction media (*n* = 4; * *p* < 0.05, ** *p* < 0.01, *** *p* < 0.001).

**Figure 3 ijms-24-15252-f003:**
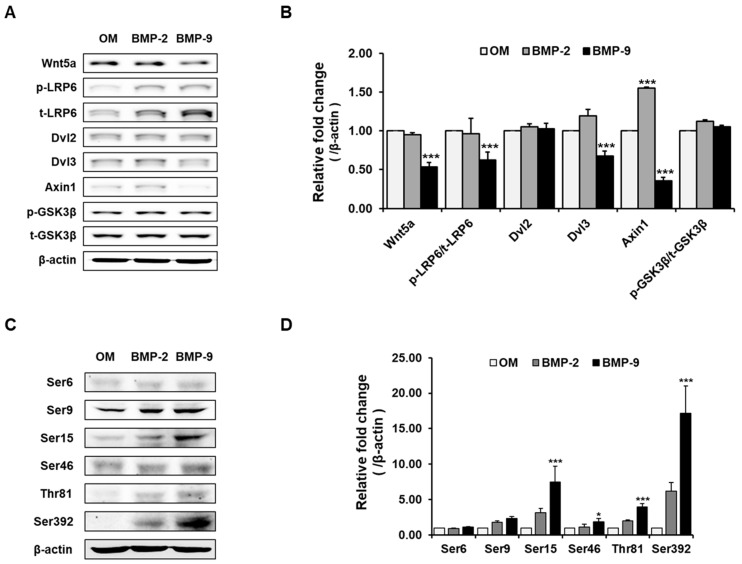
Effect of BMP-9 on Wnt signaling and the p53 factor during osteogenic differentiation. Most of the downstream factors of the Wnt signal were downregulated due to BMP-9 (**A**,**B**), and there were multiple areas where phosphorylation in specific p53 sequences significantly increased (**C**,**D**) (*n* = 3; * *p* < 0.05, *** *p* < 0.001).

**Figure 4 ijms-24-15252-f004:**
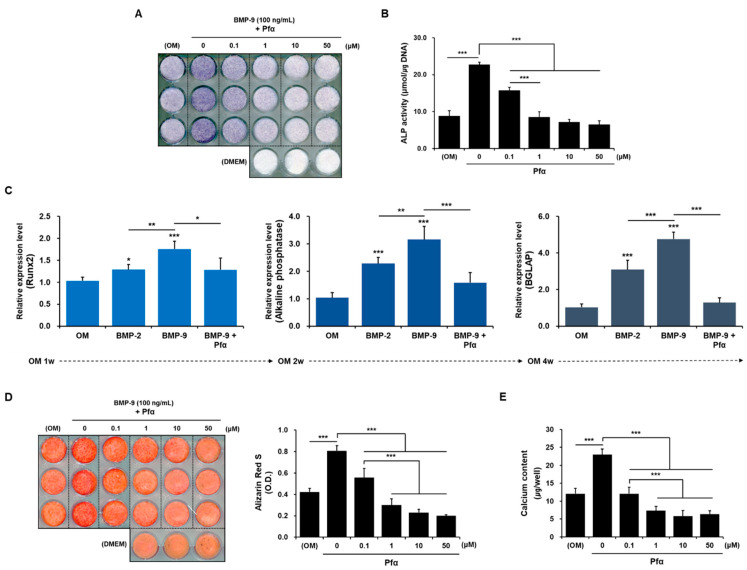
Effects of Pfα in osteogenic differentiation. A dose-dependent reduction in the degree of osteogenic differentiation due to the p53 inhibitor was confirmed through ALP staining (**A**,**B**), and the expression of mRNA levels of the osteogenic factors involved in time-dependent differentiation also decreased (**C**). The degree of calcium deposition was improved with BMP-9, then it was significantly downregulated by Pfα (**D**,**E**). (*n* = 3; * *p* < 0.05, ** *p* < 0.01, *** *p* < 0.001).

**Figure 5 ijms-24-15252-f005:**
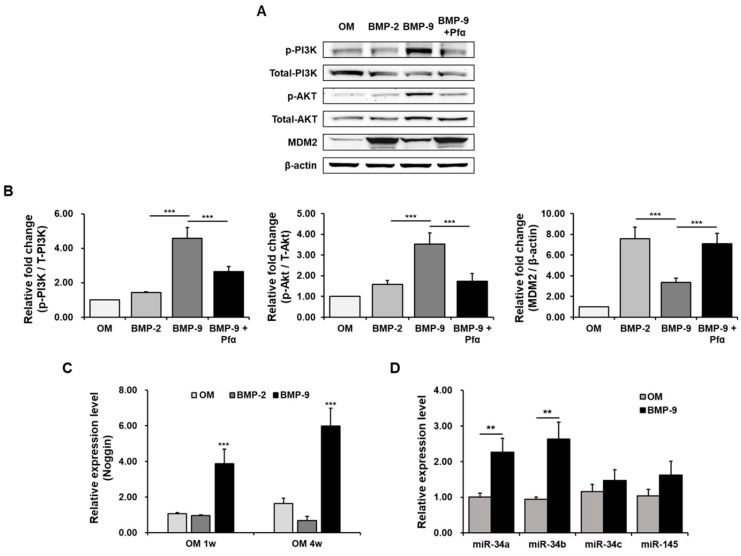
Mechanism of action of BMP-9 and p53. Western blotting and quantification graph to confirm the PI3K/AKT/MDM2 pathway (**A**,**B**), and mRNA expression data of each antagonistic factor noggin (**C**) and micro-RNA group (**D**) to confirm the function of BMP-9 and p53 (*n* = 3; ** *p* < 0.01, *** *p* < 0.001).

## Data Availability

The data that support the findings of this study are available from the corresponding author, upon reasonable request.

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
