# Peer review of "BMP-9 Improves the Osteogenic Differentiation Ability over BMP-2 through p53 Signaling In Vitro in Human Periosteum-Derived Cells"

_ijms, 2023, doi:10.3390/ijms242015252_

Round 1
Reviewer 1 Report
In the manuscript entitled "BMP-9 improves osteogenic differentiation ability over BMP-2 through p53 signaling in vitro of human periosteum-derived cells" the authors investigated the mechanism for signal transduction of BMP-9, focusing on the p53 factor. I suggest the authors address the following issues in the interests of clarity:
1. All reagents and materials should be indicated in the materials section.
2. The findings-based informative conclusion is lacking. The authors would clearly indicate the contribution of findings to the field.
3. The authors should re-read the full manuscript and reorganize chronologically the abbreviations and correct typos.
4. The language should be polished a little bit. There are several instances where the language could be improved for clarity and precision.
-
Reviewer 2 Report
**Critical Review on the Osteogenic Differentiation of hPDCs Induced by BMP-9**
**Introduction**
BMPs, or Bone Morphogenetic Proteins, play a critical role in bone formation and repair. Among them, BMP-9 has garnered attention for its pronounced osteogenesis-inducing capability, which appears distinct from other BMPs.
**Distinct Pathways of BMP-9 in Osteoblast Differentiation**
A comparative study of BMP-2 and BMP-9's influence on hPDCs, a specific type of stem cell, showed that BMP-9 significantly enhances osteoblast differentiation, more than BMP-2. The underlying mechanism, however, remains elusive. Historically, BMPs are known to activate the Runx2 protein via the MAPK signaling pathway, which is central to osteoblast differentiation. Interestingly, BMP-9 seems to diverge from this route, exhibiting low phosphorylation levels of MAPKs like JNK and p38. Instead, BMP-9's influence appears linked to an uptick in p53 phosphorylation, a known tumor suppressor and regulator of the cell cycle.
**The p53 Connection**
The role of p53, traditionally viewed as a tumor suppressor, in osteoblast differentiation has been underscored. This research confirms that p53 signaling, particularly via the PI3K/Akt/MDM2 axis, plays a pivotal role in the osteoblast differentiation process when stimulated by BMP-9. Additionally, miRNAs, which regulate mRNA degradation or translation, have emerged as crucial players in this process, with several miRNAs identified that directly or indirectly affect p53 activity.
**Experimental Methods**
The methodology detailed is robust, from the initial hPDC culture procedures to the evaluation of osteogenic phenotypes and phosphorylation array experiments. The Western blot analysis offers a comprehensive assessment of multiple signaling molecules, while the quantitative RT-PCR provides insights into the gene expression changes influenced by BMP-9.
**Implications and Limitations**
The findings highlight the promising potential of BMP-9 in clinical applications, particularly for complex bone repairs, non-union fractures, and spinal fusions. However, there are complexities to consider. Since p53 has diverse roles, including apoptosis and cell cycle regulation, the broader implications of these findings may have ramifications beyond just bone differentiation, necessitating further research.
**Conclusions**
This study illuminates the unique pathway BMP-9 takes to induce osteoblast differentiation, emphasizing its potential as a potent bone regeneration agent. The intertwining roles of the PI3K/Akt/MDM2 axis, p53 signaling, and miRNAs in this process present an intricate landscape, which, if understood more deeply, could revolutionize therapeutic approaches in bone healing. While the findings are promising, they also present a cautionary tale: the cellular mechanisms are multifaceted, and harnessing BMP-9 for therapeutic use requires a delicate balancing act to ensure desired outcomes without unwanted side effects.
Round 2
Reviewer 1 Report
The authors addressed my concerns.